# Factors Affecting Mothers’ Adherence to Home Exercise Programs Designed for Their Children with Cerebral Palsy

**DOI:** 10.3390/ijerph191710792

**Published:** 2022-08-30

**Authors:** Reem M. Alwhaibi, Asma B. Omer, Ruqaiyah Khan

**Affiliations:** 1Department of Rehabilitation Sciences, College of Health and Rehabilitation Sciences, Princess Nourah bint Abdulrahman University, P.O. Box 84428, Riyadh 11671, Saudi Arabia; 2Department of Basic Health Sciences, Deanship of Preparatory Year for the Health Colleges, Princess Nourah bint Abdulrahman University, P.O. Box 84428, Riyadh 11671, Saudi Arabia; 3Department of Public Health, Athar Institute of Health and Management Studies, New Delhi 110049, India

**Keywords:** cerebral palsy, home exercise program, mother’s compliance

## Abstract

Cerebral palsy is a common motor disorder that results in long-term impairment. The purpose of this study was to find out what factors influence Saudi mothers’ compliance with their **Children with Cerebral Palsy (C-CP**) Home Exercise Program (HEP). A self-administered online questionnaire was used to perform this qualitative research study on a group of 113 mothers who had children with CP. The study included mothers with children from birth to 12 years old who had received a HEP prescription from a physiotherapist. The measuring instrument tool was a questionnaire with two sections: demographic characteristics and a questionnaire about the parents’ adherence to the HEP. The questionnaire utilized in this study was subjected to a reliability analysis, and the derived Cronbach’s alpha was found to be 0.814 for the questionnaire (which had 17 phrases). These results imply that the questionnaire is reliable. A total of 113 responses were received, with 4 incomplete responses being eliminated. The majority of mothers (66.1%) did not follow the HEP, according to the findings of this survey. The demographics of the mothers revealed that 20–25-year-old mothers were more adherent than the other age groups. The findings of this study demonstrated that the physical therapist’s treatment of the mother influenced exercise compliance.

## 1. Introduction

Cerebral palsy (CP) is a common motor condition that causes long-term disability. It is now better recognized as a lifetime disorder, with adults with CP outnumbering children 3:1 in some countries (Australian Cerebral Palsy Register Group, 2013) [1]. Instead of being a single disease or ailment, CP is now defined as a collection of developmental disorders. A widely-accepted definition was created by an international consensus expert panel, wherein CP is: a term used to describe a group of lifelong mobility and postural abnormalities that limit activity and are caused by non-progressive anomalies in the developing fetus or infant brain [2]. CP is frequently accompanied by sensory, cognitive, communicative, and behavioral impairments, as well as epilepsy and secondary musculoskeletal problems [3]. According to Rosenbaum et al., 2007, “CP represents a collection of permanent diseases of the development of movement and posture, resulting in activity restrictions attributable to non-progressive brain abnormalities that originated during fetal or infant brain development”. The motor impairments of CP are frequently accompanied by disturbances in sensation, perception, cognition, communication, and behavior, as well as by epilepsy and secondary musculoskeletal illnesses. This definition acknowledges that determining the degree of activity restriction is a component of the CP evaluation. As well, a classification of CP with four primary components was proposed: motor abnormalities, concomitant deficits, anatomical and neuroimaging findings, and causation and timing [4].

CP prevalence estimates suggest that 1 to nearly 4 per 1000 live births or per 1000 children in recent population-based research from around the world have CP. According to 2010 estimates from the CDC’s Autism and Developmental Disabilities Monitoring (ADDM) Network, about 1 in 345 children (3 per 1000 8-year-old children) in the United States has CP [5]. Children born prematurely or with a low birth weight are more likely to develop CP. There is evidence that the prevalence of CP has decreased in several parts of the world, notably among children born at a low birth weight. The prevalence of CP in Saudi Arabia has been estimated to be 23.4–26.3 per 10,000 live births [6].

Although CP is a permanent condition, early intervention programs have been shown to increase children’s independence and quality of life, especially when applied between the ages of birth and five, when neuroplasticity is at its greatest [7,8]. As a result, family participation with the therapist is critical in maintaining the exercises during the early intervention program. Several studies have highlighted the significance of family collaboration, particularly during the implementation of a home exercise program (HEP). However, in long-term pediatric diseases, non-adherence to HEP was reported to be the primary reason for treatment failure [9].

HEP is a type of extension treatment in which a health care provider instructs parents or caregivers on how to carry out the prescribed program at home and between physical therapy appointments [10,11]. In order to get an effective outcome, the HEP often accounts for 50–80% of the overall therapy received by children with CP (C-CP) [10]. Novak (2009) claims that HEP can “improve function, parental satisfaction, objective achievement, and upper limb quality of movement when compared to having no program at all [12].” HEP is significant for rural families because it can help them save time and money while also involving parents in treatment decision-making and empowering them to choose what is best for their child [13,14].

HEP participation by parents is becoming more widespread [15]. However, adhering to the program is a significant issue [16]. Many research studies in various developmental clinical conditions have been carried out to measure parents’ adherence to HEP as well as discover the elements that may influence it. These studies were carried out in Spain [15], America [17], and Turkey [6]. Most of this research found that a lack of social support, physical therapist support and supervision, and parents’ knowledge and capacity to execute an HEP are the main factors for non-adherence [15,17,18]. For example, a study was undertaken on Brazilian mothers to better understand their views on the role of physical therapy and HEP. They discovered that the mothers were aware of the role of physical therapy and the usefulness of HEP in improving their child’s condition [19]. Saudi mothers, on the other hand, who were recruited to examine their perspective and understanding of their child’s condition, were found to be heavily reliant on Allah (God) to cure their child and to have little knowledge of the condition. Saudi mothers’ perceptions of physical therapy treatment for their children may be influenced by this, influencing the implementation of HEP [20]. Saudi women also have extra responsibilities and duties as a mother, wife, and daughter-in-law, which may contend with their disabled child’s demands [21]. Given this, a qualitative research study is needed to better understand the factors that influence Saudi mothers of children with CP’s compliance with the HEP. With this research gap in sight, the presented study was designed to determine the factors that influence Saudi mothers’ compliance with HEP for their C-CP.

## 2. Materials and Methods

### 2.1. Design

This qualitative research study was conducted through a self-administered online questionnaire on a sample of 113 mothers who have children diagnosed with CP.

### 2.2. Participants

The child disability centers were contacted to provide details of the mothers who had C-CP. After seeking their consent for their participation in the study, the questionnaire was sent to them. The inclusion/exclusion criteria were as follows:Inclusion criteria: Mothers with children aged birth to 12 years old who were given an HEP prescription from a physiotherapist.Exclusion criteria: Mothers who were unable to read or write in Arabic or who had children with other neurological disorders. In addition, the study rejected incomplete surveys.

### 2.3. Measures Included in the Questionnaires

A questionnaire was used as the measuring instrument tool and had two sections, as described below.

Demographic variables

The questions in this section were designed to collect sociodemographic information about the child, such as age (years), gender (male/female), the rehabilitation center where the child received treatment (open-ended question), and the gender of the physical therapist in charge of the child’s program. This part also included demographic questions about the mother, such as her age, marital status, educational level (primary, secondary, university, or further education), number of children, and the family’s socioeconomic situation.

2.Questionnaire related to Parents’ Adherence to HEP

This section consisted of 17 questions related to HEP adherence practices, which were adopted from previous studies [15,22]. Questions about the mother’s experience with the HEP made up the first section of the questionnaire, which included her understanding of her child’s HEP, social support, and whether she possessed the appropriate equipment. In this part, social support was assessed using a single indicator that assessed the mother’s physical support at home. It was determined by two categories and based on a five-point scale (low or high). In addition, “always” and “frequently” responses indicated strong support. Physical resource considerations included the availability of appropriate household equipment for the exercises, which were recorded as “yes/no”. Given that the role of the health professional in guiding the mother in her child’s HEP is critical, seven questions about the health professional’s engagement in the child’s condition and HEP were included. These seven questions covered three major topics: providing general information regarding the child’s condition, HEP guidelines, and exercise follow-up. All three questions were answered on a five-point frequency-based response scale with two levels of appropriate frequency—“always” or “very often”—or not. One question was related to the parent’s overall satisfaction with the physical therapy services delivered and was graded on a scale of 0 to 10 (with 0 being “totally disappointed” and 10 being “very satisfied”). The mother’s behavior regarding her adherence to HEP was the final component of the questionnaire. The frequency per week question (number of days doing the exercises in the previous week/number of days recommended per week) was used to determine it. For this component, a three-point frequency-based scale was used, i.e., “a day per week”, “three days a week”, and “seven days a week”. The physical therapist’s recommendations for the duration and repetition were assessed using a five-point frequency-based scale with the following responses: “never”, “rarely”, “sometimes”, “very often”, or “always”.

### 2.4. Translation of the Questionnaires

Using a systematic forward and backward translation technique, the questionnaire was translated into Arabic and confirmed before being submitted. The questionnaires were first translated from the English version to Arabic by a panel of three specialists who are fluent in both English and Arabic—a bilingual professional who was blind to the English version of the questionnaire’s backward translations. The backward-translated English versions of both questions were evaluated by a professional simultaneous Arabic-English interpreter.

### 2.5. Pilot Study

An initial pilot test was conducted among ten participants using the first-translated questionnaires to confirm that the questions were readily understood, and the survey length was appropriate. The researcher next conducted a face-to-face interview with interviewees to see if any of the questions were ambiguous or difficult to answer. The two first translators then edited the first-translated surveys to generate the final Arabic version, considering participants’ input and suggested modifications. The second Arabic version was re-tested with another ten people to ensure that the final version of the surveys’ instructions, questions, and response options were clear. To make the questionnaire more accessible and encourage respondents to participate, the final version was posted to an online platform (Google Forms).

### 2.6. Data Collection

The first page of the online survey contained a statement outlining the study’s details and clarifying that participation in the study was completely voluntary, that participants could withdraw at any time, and that the privacy and confidentiality of their personal information would be fully protected. Participants were authorized to complete the survey after providing informed consent, which was indicated by responding to the survey. Participants were thanked for their time and given the research group’s email address in case they wished to learn more about the study’s findings. An individual took approximately 5 min to complete the survey.

### 2.7. Data Analysis

The identity of the participants was encrypted to maintain confidentiality before running the statistics. IBM SPSS Statistics for Windows, Version 23.0, was used to analyze the data (IBM Corp., Armonk, NY, USA). The data obtained from this study was described using frequencies and percentages, mean, and standard deviation. To begin, Crosstabs was used to classify the mothers’ adherence. Second, the Chi-square test was used to evaluate the association between mothers’ adherence and demographic data, as well as to determine the relationship between mothers’ adherence and their experience. The multiple regressions test followed by Bonferroni’s correction test were used to examine the effect of the independent factors (social support, health professionals’ behavior, and mother’s experience of unsureness) on the dependent variable to validate the research hypothesis (HEP adherence). The sample used in this study was found to be slightly skewed and kurtotic for male (skewness = −0.474 ± 0.297; kurtosis = −0.844 ± 0.568) and female (skewness = 0.574 ± 0.357; kurtosis = −0.955 ± 0.702) both. However, it does not differ significantly from normality (Shapiro–Wilk test *p*-value for males = 0.200 and for females = 0.118). Hence, it was assumed that the presented sample is approximately normally-distributed in terms of skewness and kurtosis. Therefore, parametric tests were used for the analysis of the observed data.

## 3. Results

### 3.1. Reliability of the Questionnaire

The reliability of the questionnaire was conducted for the questions pertaining to the understanding of HEP, availability of equipment for HEP, assistance and support of the physiotherapist, and frequency of exercise in a week. The aim of this analysis was to assess the internal consistency of the questionnaire, hence Cronbach’s α was performed for internal consistency. The results of Cronbach’s Alpha are shown in Table 1.

The calculated Cronbach’s α was found to be 0.814 for the questionnaire (including 17 questions), these results indicate a good reliability for the questionnaire. Cronbach’s α ranges from r = 0 to 1, with r ≥ 0.7 considered as sufficiently reliable [23].

### 3.2. Outcomes of Descriptive Analysis

Recruitment was significantly lower than expected. A total of 113 responses were received, and 4 incomplete responses were excluded, which also included children with an age exceeding 12 years old. More than half of the responses were received from male C-CP (59.6%), whereas the remaining were females (40.4%). The mean age of C-CP was 69 months, which is closer to 6 years of age. Almost half of the responders (52.3%) received rehabilitation at a private facility and 47.7% of them received it at a government facility. A majority of the physiotherapists were female (70.6%), and the remaining were male (Table 2).

### 3.3. Analysis of Adherence of Mothers to Recommended HEP

For the analysis of adherence of mothers to the HEP, mothers who completed exercises with their children as recommended by the physiotherapist were considered to have good adherence, whereas others were labeled as poor. Only 33.9% (*N* = 37) of the 109 mothers followed the physiotherapist’s recommendations for child exercises, while 66.1% (*N* = 72) had poor adherence (Figure 1).

The association between mothers’ adherence to HEP and their demographic data is represented in Table 3. A statistical significance was found only in the age (Chi-square 3.307: *p*-value = 0.041), where the age group of 20–25-year-old mothers had 80% (*N* = 4) adherence. While in the other age groups, mothers’ adherence was found to be poor. However, there is no relationship between a mother’s adherence and the other demographic data, i.e., marital state, educational level, number of children, and income, because of the convergence of numbers within groups.

The association between mothers’ adherence and their everyday experience is described in Table 4. Approximately 34% of mothers had good adherence to the HEP program as they found it to fit their daily routine, with the association being significant (*p* = 0.000).

Table 5 shows a positive correlation between the adherence to HEP and predictors (mentioned in Table 4). As is evident from Table 5, R = 0.354 and R^2^= 0.125 indicate a moderate-strength correlation. R^2^ = 12.5%, which means that the independent variables (questions in Table 4) explain 12.5% of the variability of the dependent variable (adherence of mothers to the HEP). Moreover, according to Cohen’s classification, adjusted R^2^ = 0.120—as a percentage is 12.0%—which is indicative of a medium effect size. The Durbin–Watson test value was found to be 1.871, which is less than 2 points, thereby indicating a positive autocorrelation.

The multiple regression model is statistically significant, as *p* = 0.003—as shown in Table 6. This statistically significant outcome means that the addition of all the independent variables (i.e., the overall model) leads to a model that is better at predicting the dependent variable than the mean model and is a better fit. Given this, it indicates that Questions 1, 2, 3, 4, 5, and 6 presented in Table 4 are statistically significant and predicted the mothers’ adherence to HEP, F (3, 105) = 5.004, *p* < 0.005.

Table 7 demonstrates that all regression model coefficients are significant (*p* < 0.05), indicating that a mother’s sense of unsureness has a major impact on HEP adherence, with an increase of one unit in mother’s sense of unsureness resulting in a 0.537 drop in HEP adherence, while a one-unit increase in physiotherapists’ behavior corresponds to a 0.430 rise in HEP adherence. Finally, a one-unit increase in social support correlates to a 0.302 increase in HEP adherence.

## 4. Discussion

The objective of this study was to find out what factors influence Saudi mothers’ compliance with HEP for their children with CP. The outcomes of this study demonstrated that the mothers’ educational status, marital status, socioeconomic status, and the number of children in the household are all non-predictive of HEP adherence, which is consistent with earlier research [24].

According to the findings of this survey, the majority of mothers (66.1%) do not follow the HEP. The mothers who were 20–25 years old were more adherent than the other age groups, according to the demographics of the mothers. A previous study, on the other hand, found no link between the mothers’ age and HEP adherence [15,25,26]. This contradiction could be due to the study’s limited sample size, as adherent mothers outnumbered non-adherent mothers by a 4:1 ratio. In pediatric long-term illnesses, social support has been linked to adherence [18,27]. In this study of 109 mothers, the level of adherence was consistently reduced by one unit in mothers who did not have social support. This can be explained by the fact that in Saudi Arabia, the burden of raising and caring for children falls mostly on the mother, according to national customs [28]. Furthermore, contrary to the predictions of the study, the availability of home exercise equipment had no effect on adherence.

According to a 2015 study by Lillo-Navarro et al., patients’ adherence varied depending on the type of exercise, including stretching, manual skills, functional skills, postural stabilization, and sensory stimulation. When workouts were seen as having negative impacts or being too difficult, participants were less likely to be involved. The children’s responses to pain or discomfort were related to the adverse consequences, and how difficult the exercises were perceived by the parents was related to their technical proficiency. Most often, “complex exercises” were used to describe stretches and passive range-of-motion exercises. Parents who participated in these activities felt uncertain about their child’s dangers and the exercises’ end range of motion. However, the effect of differences in HEP has not been explored in the presented study [22].

### 4.1. HEP Was a Good Fit for Mothers’ Everyday Schedules

It was discovered that if the HEP fits into the mother’s daily routine, it may aid in the adherence to the exercises; however, if the HEP does not fit into their daily routine, the mothers would find it difficult to stick to their commitment. According to a study, the caretakers of impaired children experience stress and frequently struggle managing it. They discovered that caregiver stress had a significant impact on adherence [24,29]. As a result, mothers who fail to adhere to HEP due to a lack of routine may be influenced by stress.

### 4.2. Uncertainty in Mother’s Mind

It was noticed in this study that a mothers’ sense of insecurity was connected with HEP adherence in a negative way. One element impacting adherence may be a mother’s uncertainty and doubts about her ability to complete the practice. According to a study, feelings of insecurity may have an impact on adherence [18,27].

### 4.3. Behavior of a Physical Therapist

The outcome of this study revealed that the physical therapist’s behavior toward the mother has an impact on exercise adherence. As a result, the mother may dispute the necessity of HEP because of the physical therapist’s behavior. The findings of Kazdin et al. reflect the outcomes of the current study, stating that there is a bad interaction between the therapist and the family [30], thereby creating the impression to the parents that treatment is not important or overly demanding [31]. Similarly, Lillo-Navarro et al. found that a physical therapist’s training manners with parents has an impact on HEP adherence. They reported that adherence to the HEP was affected by how the physiotherapist instructed parents to develop skills for their child’s treatment. The following subthemes were discovered: increasing parents’ self-assurance in their ability to undertake exercises; assisting parents in integrating the HEP into their daily routines; offering rewards to keep adherence; and monitoring and supporting adherence [22]. The findings of this study align with the previously-reported studies and confirm that the behavior of a physiotherapist has a direct impact on parents’ adherence to HEP.

### 4.4. Limitations of the Study

The sample size gathered was small due to the restricted time available for the conduction of the study. As a result, the current study’s conclusions cannot be generalized to other situations. It’s possible that the online survey excluded mothers who didn’t have access to computers or the Internet. Since parents’ adherence practices were self-reported, replies could be influenced by the mothers’ elevated self-perception, leading to unduly optimistic comments about HEP adherence. Additionally, the study was focused only on the evaluation of mothers’ adherence to HEP, but the effect of different types of HEP was not studied. Finally, as noted in some of the omitted comments, several of the mothers were unfamiliar with the medical term “cerebral palsy” due to a misunderstanding of its meaning.

### 4.5. Recommendations

Understanding the factors that influence HEP adherence is important because it can enhance health care and patient outcomes, thus improving patients’ quality of life. It may also assist the physical therapist in identifying parents who have risk factors for poor adherence and managing them from the start. The physical therapist’s behavior was one of the elements affecting adherence, according to our data. As a result, the therapist should evaluate how he or she interacts with the child’s caregiver by delivering a written HEP instruction tailored to everyone, complete with pictures. The therapist should also describe and show the program to the caregiver using the caregiver’s child, and then have the caregiver demonstrate to the therapist for the validation of comprehension of the required task. The caregiver’s confidence in performing the exercises may improve because of this practice.

Awareness workshops for mothers of C-CP who are being recommended to participate in HEP might be devised to raise their awareness of the importance of implementing the activities at home. The programs may cover the benefits and drawbacks of implementation, as well as common roadblocks and how to overcome them. Hospitals, rehabilitation institutions, and local government health centers can all host the activities.

## 5. Conclusions

In an attempt to identify the factors influencing Saudi mothers’ adherence to HEP designed for their C-CP, the presented study found that adherence behaviors to HEP and the mothers’ potential were negatively associated. The reason could be a mother’s sense of unsureness about the effect of her child’s HEP; however, HEP was positively associated with health professionals’ behavior more than social support effects, which was found to be the lowest factor.

## Figures and Tables

**Figure 1 ijerph-19-10792-f001:**
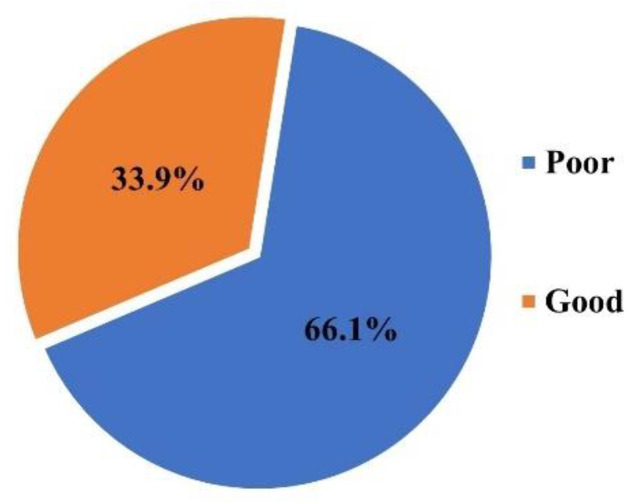
The percentage of poor and good adherence to the HEP by the mothers of C-CP.

**Table 1 ijerph-19-10792-t001:** Cronbach’s α Reliability.

	No. of Questions	Cronbach’s Alpha
Questionnaire	17	0.814

**Table 2 ijerph-19-10792-t002:** Demographic Data of Children (*N* = 109).

	Categories	*N* (%)
Gender of child	Male	65 (59.6%)
	Female	44 (40.4%)
Age of child	Mean = 69 Month, St. D = 36.50	
The type of hospital in which the child receives rehabilitation	Governmental	52 (47.7%)
	Private	57 (52.3%)
Gender of Physiotherapist responsible for the child	Male	32 (29.4%)
	Female	77 (70.6%)

**Table 3 ijerph-19-10792-t003:** Chi-square test results for analyzing association between mothers’ demographics and their adherence to HEP.

		Mothers’ Adherence		Total	Chi-square = 3.307 *p*-value = 0.041 *
		Poor	Good		
Age	20–25 Years	1	4	5	
	26–30 Years	16	9	25	
	31–35 Years	22	7	29	
	36–40 Years	24	12	36	
	More than 40 years	8	6	14	
		Mothers’ Adherence		Total	Chi-square = 1.051 *p*-value = 0.591
		Poor	Good		
Status	Married	66	35	101	
	Divorced	5	1	6	
	Widow	1	1	2	
		Mothers’ Adherence		Total	Chi-square = 3.363 *p*-value = 0.339
		Poor	Good		
Education	Less than Secondary	10	6	16	
	Secondary	22	16	38	
	Universal	37	15	52	
	Above Universal	3	0	3	
		Mothers’ Adherence		Total	Chi-square = 10.611 *p*-value = 0.231
		Poor	Good		
Children	0–2	24	14	38	
	3–5	41	17	58	
	6–10	7	6	13	
		Mothers’ Adherence		Total	Chi-square = 4.164 *p*-value = 0.384
		Poor	Good		
Income	Less than 5000	13	10	23	
	5000–10,000	27	8	35	

*: significant at 0.05

**Table 4 ijerph-19-10792-t004:** Relation between mothers’ adherence and their experience.

		Mothers’ Adherence		Total	Chi-square = 22.320 *p*-value = 0.000 **
		Poor	Good		
1. Does the home program fit your daily routine?	Rarely	11	0	11	
	Sometimes	52	18	70	
	Often	6	11	17	
	Always	3	8	11	
		Mothers’ Adherence		Total	Chi-square = 14.347 *p*-value = 0.006 **
		Poor	Good		
2. I understand my child’s home program	Strongly disagree	2	0	2	
	Disagree	12	1	13	
	Undecided	10	3	13	
	Agree	41	20	61	
	Strongly agree	7	13	20	
		Mothers’ Adherence		Total	Chi-square = 21.136 *p*-value = 0.000 **
		Poor	Good		
3. I am skillful in carrying out the home program	Strongly disagree	5	0	5	
	Disagree	29	3	32	
	Undecided	9	7	16	
	Agree	27	20	47	
	Strongly agree	2	7	9	
		Mothers’ Adherence		Total	Chi-square = 15.869 *p*-value = 0.103
		Poor	Good		
4. How confident are you about performing the home program	<5	33	7	40	
	5 and above	49	30	69	
		Mothers’ Adherence		Total	Chi-square = 5.484 *p*-value = 0.241
		Poor	Good		
5. My partner supports me at home	Never	16	5	21	
	Rarely	16	5	21	
	Sometimes	26	17	43	
	Often	11	5	16	
	Always	3	5	8	
		Mothers’ Adherence		Total	Chi-square = 2.808 *p*-value = 0.104
		Poor	Good		
6. I have the equipment required to do the exercises at home	No	47	18	65	
	Yes	25	19	44	

**: significant at 0.01.

**Table 5 ijerph-19-10792-t005:** Model Summary.

Model	R	R^2^	Adjusted R^2^	Std. Error of the Estimate	Durbin-Datson Test
1	0.354 ^a^	0.125	0.100	1.62340	1.871

^a^ Predictors: (Constant), mother’s sense of unsureness, health professionals’ behavior, social support.

**Table 6 ijerph-19-10792-t006:** ANOVA ^a^.

Model		Sum of Squares	df	Mean Square	F	Sig.
1	Regression	39.564	3	13.188	5.004	0.003 ^b^
	Residual	276.721	105	2.635		
	Total	316.284	108			

^a^ Dependent Variable: HEP adherence, ^b^ Predictors: (Constant), mother’s sense of unsureness, health professionals’ behavior, social support.

**Table 7 ijerph-19-10792-t007:** Multiple regression results for HEP adherence ^a^.

Model		Unstandardized Coefficients		Standardized Coefficients	t	Sig.
		B	Std. Error	Beta		
1	(Constant)	2.759	0.648		4.255	0.000
	Social support	0.302	0.242	0.156	1.249	0.041
	Health professionals’ behavior	0.430	0.190	0.268	2.265	0.026
	Mother’s sense of unsureness	−0.537	0.179	−0.306	−3.003	0.003

Hence, it can be stated that the multiple regression model is statistically significant and predicted the adherence of mothers to HEP, F (3, 105) = 5.004, *p* < 0.005, adj. R^2^ = 0.100. All the six variable (in Table 4) added significantly to the prediction, *p* < 0.05. ^a^ Dependent Variable: HEP adherence.

## Data Availability

The datasets used and/or analyzed during the current study are available from the corresponding author on reasonable request.

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
