# Peer review of "Factors Affecting Mothers’ Adherence to Home Exercise Programs Designed for Their Children with Cerebral Palsy"

_ijerph, 2022, doi:10.3390/ijerph191710792_

Round 1

Reviewer 1 Report

In this paper,  the authors created, validated with an interview in person and then tested an online questionnaires about mothers’ adherence to their child therapy (Home Exercise Program), in a population of 113 mothers of children with a diagnosis of cerebral palsy from Saudi Arabia. Results showed, among others, that mothers aged 20-25 year were more compliant than older mothers, and that when the child’s therapy fitted with their daily routine mother’s adherence to the therapy was higher.  Results also highlighted the importance of an alliance between mothers and therapists.  

I think that the rationale of the study is concrete and interesting and the results are potentially interesting as well, especially considering the cultural framework in which the study has been performed. However, there are several adjustments, mainly in the methods and results, that need to be addressed to avoid confusion, which the case in the current state of the manuscript.  Methods and results are indeed not always clear to me. Therefore, even though potentially interesting, results can result difficult to be interpreted in some occasions.  Also the abstract and the main text are difficult to understand at different points, and I believe there is an overall need for a moderate editing of English language and style. A more appropriate style and use of English would definitely help the fluency thus greatly benefitting  the manuscript. 

Here are some specific major and minor comments/suggestions for the authors:

Abstract

-        The abstract could be more concise. I do not think there is need to include so many details in it. Authors should describe more generally methods and results from the study, and briefly comment them.

-        The acronym C-CP should be "Children with cerebral palsy (C-CP)" as it is the first time authors use it.

Introduction

-        I would suggest to use the definition of Cerebral Palsy by Rosenbaum et al., 2007 (see reference below), which the authors reference to already in the manuscript. 

Rosenbaum, P., Paneth, N., Leviton, A., Goldstein, M., & Bax, M. (2007). A report: the definition and classification of cerebral palsy. Developmental Medicine & Child Neurology49, 8–14. 

-        I would smooth the statement “cerebral palsy cannot be cured” at line 52 rephrasing it with something like “Although cerebral palsy is a permanent condition”.

-        I would talk of “clinical conditions” rather than “problems” at line 71.

-       What do the authors mean by “physical therapist conduct” at line 74?please clarify this point. 

Methods and Results 

-        In the Participants section (line 94) I would just state “Inclusion/Exclusion criteria were as follows:” 

-        At line 102: the word “under” should be “below”.

-   At lines 111-112 authors should clarify what the probable variable they refer to are. And, always in in relation to this comment, I would suggest to include the actual questionnaire administered in the manuscript or in supplementary materials, to avoid any confusion about the results as well.

-        It is not clear to me what kind of method used to assess reliability of the questionnaire as a tool of investigation. Authors state that used Cronbach’s Alpha, but the actual comparisons they made are not clear to me. Methods about reliability should be better and further described. What did actually the authors compared to calculate Cronbach’s Alpha?

-        Did the authors perform a power analysis (a priori or a posteriori) to establish the minimum sample size of mothers? They state at different points in the manuscript that the number of responsive mothers was lower than expected but this point is not clear to me and should be clarified. The minimum number of subjects required should be calculated through a power analysis and declared in the main text.

-      Section 3.3. Analysis of adherence of mothers to recommended HEP: Authors should specify that those data were extrapolated from the questionnaires (see also my previous comment about the confusion due to not having the questionnaire available).

-        The paragraph about Table 5 should be revised. It is not clear to me which one among the predictors did positively correlate with adherence to HEP. Also, the p value/s should be reported in Table 5.

-        In general, did the authors apply any correction for multiple comparisons? If yes, please specify what kind of correction was applied. If not, authors should apply it to the p values. 

Discussion

I appreciate that the authors clearly indicated some of the limitations of the study. I would, though, remove the sentence at lines 274-275 “Because of the high demand for childcare, the length  of the questionnaire may have deterred mothers of children with cerebral palsy from completing it.”. I think that the length of the questionnaire, which is 5-minute-long, would not affect mothers’ time and effort in filling it so much.

Author Response

Response to Reviewer 1 Comments

Point 1: The abstract could be more concise. I do not think there is need to include so many details in it. Authors should describe more generally methods and results from the study, and briefly comment them.

Response: Abstract has been modified as suggested.

Point 2: The acronym C-CP should be "Children with cerebral palsy (C-CP)" as it is the first-time authors use it.

Response: The correction has been made as suggested by the reviewer.

Point 3: I would suggest using the definition of Cerebral Palsy by Rosenbaum et al., 2007 (see reference below), which the authors reference to already in the manuscript.

Response: The suggestion has been included.

Point 4: I would smooth the statement “cerebral palsy cannot be cured” at line 52 rephrasing it with something like “Although cerebral palsy is a permanent condition”.

Response: The suggestion has been accepted and changes were made.

Point 5: I would talk of “clinical conditions” rather than “problems” at line 71. What do the authors mean by “physical therapist conduct” at line 74? please clarify this point.

Response: The suggestion has been accepted and changes were made. By the word conduct the authors meant to describe the behavior of physical therapist.

Point 6: In the Participants section (line 94) I would just state “Inclusion/Exclusion criteria were as follows:”

Response: The suggestion has been accepted and changes were made.

Point 7: At line 102: the word “under” should be “below”.

Response: The suggestion has been accepted and changes were made.

Point 8: At lines 111-112 authors should clarify what the probable variable they refer to are. And, always in in relation to this comment, I would suggest to include the actual questionnaire administered in the manuscript or in supplementary materials, to avoid any confusion about the results as well.

Response: The suggestion has been accepted and the questionnaire has been attached as supplementary material.

Point 9: It is not clear to me what kind of method used to assess reliability of the questionnaire as a tool of investigation. Authors state that used Cronbach’s Alpha, but the actual comparisons they made are not clear to me. Methods about reliability should be better and further described. What did actually the authors compared to calculate Cronbach’s Alpha?

Response: The authors apologize for the misunderstanding, Reliability analysis of the questionnaire was conducted for the questionnaire questions pertaining to the understanding of HEP, availability of equipment for HEP, assistance and support of the physiotherapist, frequency of exercise in a week. The aim of this analysis was to assess the internal consistency of the questionnaire, hence Cronbach’s α was performed for internal consistency.

Also, the questionnaire has been attached for further reference as supplementary material.

Point 10: Did the authors perform a power analysis (a priori or a posteriori) to establish the minimum sample size of mothers? They state at different points in the manuscript that the number of responsive mothers was lower than expected but this point is not clear to me and should be clarified. The minimum number of subjects required should be calculated through a power analysis and declared in the main text.

Response: Due to statistical incompetency, we did not calculate the sample size. Moreover, the sample was based on the availability of the mothers to complete the survey.

Point 11: Section 3.3. Analysis of adherence of mothers to recommended HEP: Authors should specify that those data were extrapolated from the questionnaires (see also my previous comment about the confusion due to not having the questionnaire available).

Response: The suggestions has been considered and the questionnaire was attached. The extrapolation is based on the questions mentioned in the Table no. 4. We have added more explanation to the paragraph. The authors request the reviewer to suggest any further improvement, if needed.

Point 12: The paragraph about Table 5 should be revised. It is not clear to me which one among the predictors did positively correlate with adherence to HEP. Also, the p value/s should be reported in Table 5.

Response: The paragraph has been revised and explained in detail. The Durbin-Watson Test values were added to report autocorrelation.

Point 13: In general, did the authors apply any correction for multiple comparisons? If yes, please specify what kind of correction was applied. If not, authors should apply it to the p values.

Response: Yes, we applied the correction, it was Bonferroni’s correction.

Point 14: I appreciate that the authors clearly indicated some of the limitations of the study. I would, though, remove the sentence at lines 274-275 “Because of the high demand for childcare, the length of the questionnaire may have deterred mothers of children with cerebral palsy from completing it.”. I think that the length of the questionnaire, which is 5-minute-long, would not affect mothers’ time and effort in filling it so much.

Response:  The suggestion has been accepted and changes were made.

Reviewer 2 Report

Thank-you for completing this interesting study exploring factors influencing adherence of Saudi mothers of children with CP to PT home programs.

Overall, the paper is well written and clear. I have only a few comments on language – Please use ‘child’ instead of ‘kid’. ‘Kid’ is slang and not appropriate for an academic paper – see lines 68 and 289.  Also, line 102 – I think you mean ‘as described below’ rather than ‘as described under’. Line 160 should say ‘responding to the survey’ rather than ‘responding the survey’.

In addition, I have a few comments and recommendations:

Abstract:

Is a little long and lacks focus. It would be more helpful to provide more detail on the results and implications, and only necessary detail on the methods.

Introduction:

Generally, well written with appropriate references. Please use CP consistently once the abbreviation has been introduced instead of sometimes using cerebral palsy. A little more detail on what the HEP may entail may be helpful – see additional comments under methods section below.

Methods:

I notice that you do not mention ethics review – was the study reviewed by an ethics board? If an online survey does not require ethics review in Saudi Arabia, then you should state this.

How were subjects recruited?

You state that participant data was anonymized prior to analysis – but were the online surveys matched in any way to the IP address or participants computer – usually this is a concern for ethics boards...

No detail is provided on the nature of the HEP – it is only referred to as exercises. This makes me wonder if the programs are based on child and family selected goals and preferences – or integrated into functional activities and routines designed to enhance the child’s participation with family and friends (as is current best practice). If the HEP consists of outdated exercises, that have not been shown to lead to improved function - this could also contribute to lack of adherence. Was there any difference between the nature of the HEP for different participants?

Results:

Given the very small numbers in the sub-group of mothers aged 20-25, I do not think it is appropriate to draw any conclusions regarding a difference in adherence.  Also, there is overlap between the age groups e.g. 20-25, 25-30, 30-35 etc. Why were 5-year age bands chosen?  Did you think there would be a difference between those – and how was this justified? Where was the split made – should it be 20-25, 26-30, 31-35 etc? Was there a difference if mothers were changed into the next age group or the groups were combined – e.g. 20-29, 30-39, 40 and over. Was there a difference if you kept age as a continuous variable and analyzed in your regression model?

Discussion:

More detail on the fit between the HEP and the mother’s schedule would be helpful – and perhaps discussion about how the program could be integrated into functional activities and routines to increase participation for the child.

Behaviour of the PT is a key point in this study – and should be expanded. The relationship of the PT to the mother, and whether they were collaborating on the development of the HEP is important.

Appendix – adding the full survey as an appendix would increase clarity for the reader. For example, I am not entirely sure how the PTs behaviour was rated – and this is a key point for your discussion

Author Response

Response to Reviewer 2 Comments

Point 1: Overall, the paper is well written and clear. I have only a few comments on language – Please use ‘child’ instead of ‘kid’. ‘Kid’ is slang and not appropriate for an academic paper – see lines 68 and 289.  Also, line 102 – I think you mean ‘as described below’ rather than ‘as described under’. Line 160 should say ‘responding to the survey’ rather than ‘responding the survey’.

Response: The suggestion has been accepted and changes were made.

Point 2: Abstract is a little long and lacks focus. It would be more helpful to provide more detail on the results and implications, and only necessary detail on the methods.

Response: Abstract has been modified as suggested.

Point 3: Introduction is well written with appropriate references. Please use CP consistently once the abbreviation has been introduced instead of sometimes using cerebral palsy. A little more detail on what the HEP may entail may be helpful.

Response: The suggestion has been accepted and changes were made.

Point 4: I notice that you do not mention ethics review – was the study reviewed by an ethics board? If an online survey does not require ethics review in Saudi Arabia, then you should state this.

How were subjects recruited?

You state that participant data was anonymized prior to analysis – but were the online surveys matched in any way to the IP address or participants computer – usually this is a concern for ethics boards...

Response: The suggestion has been accepted and changes were made. The ethics review is available at line number 324. The recruitment process has been described in participants sections at line 104. Each time a user accessed the eligibility screening questionnaire, the survey program logged the user's IP address. All data were treated as protected health information and were maintained on a secure server in a password-protected file with additional identifying information about the participant. This file was kept separate from the file containing de-identified participant responses.

Point 5: No detail is provided on the nature of the HEP – it is only referred to as exercises. This makes me wonder if the programs are based on child and family selected goals and preferences – or integrated into functional activities and routines designed to enhance the child’s participation with family and friends (as is current best practice). If the HEP consists of outdated exercises, that have not been shown to lead to improved function - this could also contribute to lack of adherence. Was there any difference between the nature of the HEP for different participants?

Response: The reviewer has raised a genuine concern about the differences in the HEP, but our study was mainly focused on adherence to HEP only as we have selected a population of mothers with children aged birth to 12 years old.

Point 6: Given the very small numbers in the sub-group of mothers aged 20-25, I do not think it is appropriate to draw any conclusions regarding a difference in adherence.  Also, there is overlap between the age groups e.g. 20-25, 25-30, 30-35 etc. Why were 5-year age bands chosen?  Did you think there would be a difference between those – and how was this justified? Where was the split made – should it be 20-25, 26-30, 31-35 etc? Was there a difference if mothers were changed into the next age group or the groups were combined – e.g. 20-29, 30-39, 40 and over. Was there a difference if you kept age as a continuous variable and analyzed in your regression model?

Response: We acknowledge that a band of 20-29 and 30-39 and likewise is a better and standard way of selecting, however, the authors  wend for selection of 5 year age band as early age marriages are quite common in the KSA and the authors assumed that it will  show better results.

Point 7: More detail on the fit between the HEP and the mother’s schedule would be helpful – and perhaps discussion about how the program could be integrated into functional activities and routines to increase participation for the child.

Response: The authors are thankful to the reviewer for raising the point, but, this research had limitations and we were only looking for mother’s adherence to the HEP and the factors affecting it.

Point 8: Behaviour of the PT is a key point in this study – and should be expanded. The relationship of the PT to the mother, and whether they were collaborating on the development of the HEP is important.

Response: To be precise, the PT behavior was one of the factors which was used to find its impact to the adherence of mothers to the HEP.

Point 9: Appendix – adding the full survey as an appendix would increase clarity for the reader. For example, I am not entirely sure how the PTs behaviour was rated – and this is a key point for your discussion

Response: The full survey questionnaire was added as suggested by the reviewer.

Round 2

Reviewer 2 Report

Thank-you for addressing my previous comments and questions. The abstract has been much improved and the slang and most English language issues corrected.

As regards to ethics review - I do not see any detail about the ethics at line 324 - this is the Conclusion. There is a title 'Ethics review board statement' at the end - but no comment.

Methods - like the other reviewer, I have some concerns about the methodology and the lack of reporting detail. Adding the questionnaire to the appendix has increased clarity - but now being aware of the ordinal nature of all the data collected, I wonder whether the ANOVA is an appropriate analysis - and whether an ordinal regression would be more appropriate - I would suggest a review by someone with more expertise in statistical analysis - and the addition of more explanation/justification to the manuscript.  For example, were parametric analyses appropriate - were data normally distributed? Or, would non-parametric tests be more appropriate? 

The other reviewer asked questions regarding the analyses used - and although some detail has been added in the response to the reviewer, information about the multiple analyses and bonferroni correction have not been added to the manuscript.

Limitations - the lack of information on the HEP is a limitation and should be added to that section. 

More discussion about potential differences in HEP and therapist relationship/interaction could be added to the discussion

Author Response

Response to Reviewer 2 Comments

Authors would like to express their gratitude to the reviewer for guiding us in finalizing the manuscript and in shaping the manuscript towards perfection.

Point 1: Thank-you for addressing my previous comments and questions. The abstract has been much improved, and the slang and most English language issues corrected.

Response: Authors are thankful to the reviewer for his positive feedback and his expert guidance. The correction is a result of his genuine feedback.

Point 2: As regards to ethics review - I do not see any detail about the ethics at line 324 - this is the Conclusion. There is a title 'Ethics review board statement' at the end - but no comment.

Response: The authors apologize for the mistake; we had it in consideration, but we missed to put it in place. The rectification has been made with following addition to the manuscript from line 374-382.

Ethical Considerations: All ethical approvals were obtained from the Departmental scientific sub-committee, Research Ethical Committee of College of Health and Rehabilitation Sciences and Institutional Review Board (IRB Log number: 18-0264; IRB Registration number with KACST, KSA: H-01-R-059) of Princess Nourah bint Abdulrahman University before conducting the study. Prior to contacting the mothers, internal clearances were also acquired from the participating centers. By giving all recruited mothers consent forms to sign prior to being questioned, researchers were able to confirm that they were all participating voluntarily. Finally, participant confidentiality was upheld during all phases of the research process, including recruiting, data collection, analysis, and reporting.”

Point 3: Methods - like the other reviewer, I have some concerns about the methodology and the lack of reporting detail. Adding the questionnaire to the appendix has increased clarity - but now being aware of the ordinal nature of all the data collected, I wonder whether the ANOVA is an appropriate analysis - and whether an ordinal regression would be more appropriate - I would suggest a review by someone with more expertise in statistical analysis - and the addition of more explanation/justification to the manuscript.  For example, were parametric analyses appropriate - were data normally distributed? Or, would non-parametric tests be more appropriate?

Response: The authors are thankful for the question, we would like to inform that the ANOVA is a test following the multiple regression, it is an output of the SPSS, which inform us whether the regression model results in a statistically significant prediction of the dependent variables. Following which, in our study we have mentioned the following in our manuscript:

“The multiple regression model is statistically significant as p=0.003, as shown in Table 6. This statistically significant out-come means that addition of all the independent variables (i.e., the overall model) leads to a model that is better at predicting the dependent variable than the mean model and is a better fit. So, it indicates that the questions (1, 2, 3, 4, 5 and 6) presented in the Table 4 statistically significantly predicted mother’s adherence to HEP, F (3, 105) =5.004, p<0.005.”.

The above-mentioned paragraph is mentioned between 257-264.

Moreover, regarding the normality of obtained sample in the presented study, following addition has been made and a reason has been given for the use of parametric tests (see line 189-195).

“The sample used in this study was found to be slightly skewed and kurtotic for male (skewness= -0.474±0.297; kurtosis= -0.844±0.568) and female (skewness= 0.574±0.357; kurtosis= -0.955±0.702) both. However, it does not differ significantly from normality (Shapiro-wilk test P value for males=0.200 and for females=0.118). Hence, it was assumed that the presented sample is approximately normally distributed in terms of skewness and kurtosis. Therefore, parametric tests were used for analysis of the observed data.”

Point 4: The other reviewer asked questions regarding the analyses used - and although some detail has been added in the response to the reviewer, information about the multiple analyses and bonferroni correction have not been added to the manuscript.

Response: The authors have added more information about multiple analyses (257-264; 273-275) and Bonferroni correction has been included in the manuscript (line number 186).

Point 5: Limitations - the lack of information on the HEP is a limitation and should be added to that section.

Response: The authors are grateful for the suggestion; it has been added to line 339-340.

Point 6: More discussion about potential differences in HEP and therapist relationship/interaction could be added to the discussion

Response: We thank the reviewer for the valuable suggestions. Discussion about potential differences in HEP is included line 294-303; therapist relationship/interaction has been added from line 324-330.
